# Corrosion Behavior in RC Member with Different Cover Depths under Cyclic Chloride Ingress Conditions for 2 Years

Kwang-Myong Lee [1], Yong-Sik Yoon [2], Keun-Hyeok Yang [3], Bong-Young Yoo [4] and Seung-Jun Kwon [2,*]

1   Department of Civil, Architectural and Environmental System Engineering, Sungkyunkwan University, Suwon 16419, Republic of Korea
2   Department of Civil and Environmental Engineering, Hannam University, Daejeon 34430, Republic of Korea
3   Department of Architectural Engineering, Kyonggi University, Suwon 16227, Republic of Korea
4   Department of Materials Science and Chemical Engineering, Hanyang University, Ansan 15588, Republic of Korea
*   Correspondence: jjuni98@hannam.ac.kr; Tel.: +82-42-629-8020

**Abstract:** Concrete structures are considered as durable construction material, but corrosion of the embedded steel reinforcement occurs under chloride exposure as concrete has porous properties. Herein, a cyclic drying–wetting test was performed for two years using saltwater to accelerate steel corrosion in a reinforced concrete (RC) member. The open-circuit potential (OCP) was measured using a newly developed and replaceable agar sensor. The corrosion potential was measured considering the chloride concentration, water-to-cement ($w/c$) ratio, and cover depth at three levels. Furthermore, its relationships with influential parameters were evaluated using averaged OCP results. The measured OCP showed a linear relationship with the cover depth, and this tendency was more distinct with increasing retention period and higher chloride concentration. For the highest $w/c$ ratio (0.6), values below −100 mV were monitored after only six months regardless of the cover depth, and values below the critical potential level (−450 mV) were evaluated at lower cover depths (30 and 45 mm). The results of regression analysis considering the exposure environment showed a clear relationship in the case of high chloride concentration (7.0%). A linear relationship between cover depth and OCP was derived with a reasonable determination coefficient ranging from 0.614 to 0.771.

**Keywords:** chloride ingress; corrosion monitoring; OCP; cover depth; RC member





## 1. Introduction

Concrete has been exposed to various exposure environments as a durable construction material, and long service life is usually required for reinforced concrete (RC) structures. RC structures are vulnerable to tensile stress; thus, they are reinforced with steel reinforcement or tendons to support their loads. Steel reinforcement is known to be protected from corrosion due to the high alkalinity of concrete; however, depending on the exposure environment, corrosion frequently occurs in embedded steel through the penetration of chloride and sulfate ions, pH reduction caused by carbonation, and the periodic repetition of drying and wetting by moisture [1–4]. Steel reinforcement and tendons in RC members are mainly resistant to tensile forces. When corrosion occurs, the bond strength increases owing to the swelling effect of steel reinforcement up to a corrosion ratio of 3–5%. However, the slip of steel reinforcement occurs and causes a serious safety problem of the structure when the critical corrosion amount is exceeded [5]. It is crucial to detect corrosion and evaluate its behavior as early as possible as the corrosion of embedded reinforcements causes enormous economic loss and rapidly propagates over time [6,7].

Many studies have been conducted on modeling and control mechanisms for the corrosion of steel by chloride attack and carbonation. Furthermore, research on corrosion prediction and service life analysis has recently been conducted along with BIM (building information modeling) or engineering uncertainties modeling [8–10]. However,

studies on corrosion detection for RC structures have many limitations because of the difference in corrosion behavior in a laboratory experiment and that of an actual structure, the non-homogeneity of the concrete material, and the influence of local conditions. The nondestructive technique (NDT) has been used in many fields, as it evaluates the performance of a structure by collecting the representative index without destroying it. It has also been applied for evaluating the corrosion feasibility of steel reinforcement. As reported in [11], the results from electrochemical sensors were summarized as OCP (open-circuit potential), surface potential, concrete resistivity, noise analysis, and galvanic current. For OCP measurement, not only conventional reference electrodes such as SCE and CSE but also new material such as activated titanium and $MnO_2$ were attempted for measurement [12–14]. Concrete resistivity is a special characteristic of the material that can permit the amount of carrying current. It shows qualitive results of corrosion feasibility but is very useful for field assessment [15,16]. Polarization resistance and EIS (Electrochemical Impedance Spectroscopy) measurement can provide more accurate and quantitative results such as corrosion current [17]. For galvanic current measurement, many studies have been performed considering environmental conditions and residual chloride contents [18,19]. Measurements of influencing parameters on corrosion have been carried out such as pH [20], oxygen [12], and chloride content [21,22]. In the previous research [11], the sensor mechanism and classification of the measurement method were well explained.

Monitoring sensors that are conventionally installed on embedded steel reinforcement during the construction stage can evaluate the relatively quantitative corrosion behavior (potential and corrosion density) through measuring the potential difference among the reference electrode (RE), working electrode (WE), and counter electrode (CE) connected to the embedded steel reinforcement. Although the embedded steel reinforcement and the related measuring electrodes are safe, significant misleading errors from actual corrosion potentials may occur when the electrolyte life of the embedded sensors expires or the internal medium is supersaturated by high-concentration ion penetration.

In this study, a sensor that can be replaced and calibrated outside was fabricated using agar as an ion transfer material. RC specimens were prepared based on Type 1 ordinary Portland cement (OPC), and the changing OCP was analyzed for two years considering durability design parameters such as $w/c$ ratio, cover depth, and exterior chloride concentration. Evaluation of the corrosion behavior through comparison with measured OCP data and the durability design parameters is important as it can provide a relative corrosion risk among various design parameters. For this reason, changes in the measured values were analyzed for concrete samples with three different $w/c$ ratios subjected to repeated drying and wetting for two years, and the correlation was evaluated through the averaging method.

## 2. Corrosion Mechanism and Its Detection Techniques

### 2.1. Corrosion Mechanism in Steel in RC and PSC

Pore water in concrete has various ions such as Na (OH) and $CaCl_2$, and the pH in pore water is over 12.0 due to the high alkalinity of cement hydrates. Under high-alkalinity conditions, the surface of steel can be protected by a passive film from external acid and ions. However, steel corrosion easily occurs owing to the consumption of hydrates, decrease in pH, and intrusion of halogen ions. Among the anions, the chloride ion ($Cl^-$) easily penetrates into concrete and yields local corrosion, so-called pitting. The anodic and cathodic reactions can be expressed as Equations (1)–(3) [4,23].

$$Fe \ \rightarrow \ Fe^{2+} + 2e^- \ \text{(Anodic reaction)} \tag{1}$$

$$H_2O + \frac{1}{2}O_2 + 2e^- \rightarrow 2(OH)^- \ \text{(Cathodic reaction)} \tag{2}$$

$$Fe + \frac{1}{2}O_2 + H_2O \rightarrow Fe(OH)_2 \ \text{(Overall reaction)} \tag{3}$$

When oxygen is sufficient in the anodic areas, ferrous hydroxide, $Fe(OH)_2$, can be further oxidized to other corrosion products, and this is accompanied by increasing volume, expressed as Equations (4)–(6).

$$\text{Ferrous hydroxide}: Fe^{2+} + 2OH^- \rightarrow Fe(OH)_2 \tag{4}$$

$$\text{Ferric hydroxide}: 4Fe(OH)_2 + O_2 + 2H_2O \rightarrow 4Fe(OH)_3 \tag{5}$$

$$\text{Hydrated ferric oxide (rust)}: 2Fe(OH)_3 \rightarrow Fe_2O_3{\cdot}H_2O + 2H_2O \tag{6}$$

Volume swelling at the steel is reported to range from a factor of two to ten, and this leads to cracking and spalling of concrete. In addition to the corrosion mechanism mentioned above, crevice corrosion and macro-corrosion can easily occur in steel reinforcement or PS (Pre-Stressing) tendons close to corrosion positions [23,24]. For PS tendons, in particular, a complex corrosion mechanism occurs as the macro-cathode area and small macro-anode areas form by the cavities in the duct or grout quality degradation unlike RC structures. There are various causes of corrosion, such as sulfur ions in cement, the movement of deicing agents and moisture from the upper slab, and the influence of chalky grout. Macro–micro-complex corrosion mechanisms simultaneously occur as the strands are very adjacent to each other [23]. Figure 1 shows schematics of various corrosion mechanisms.

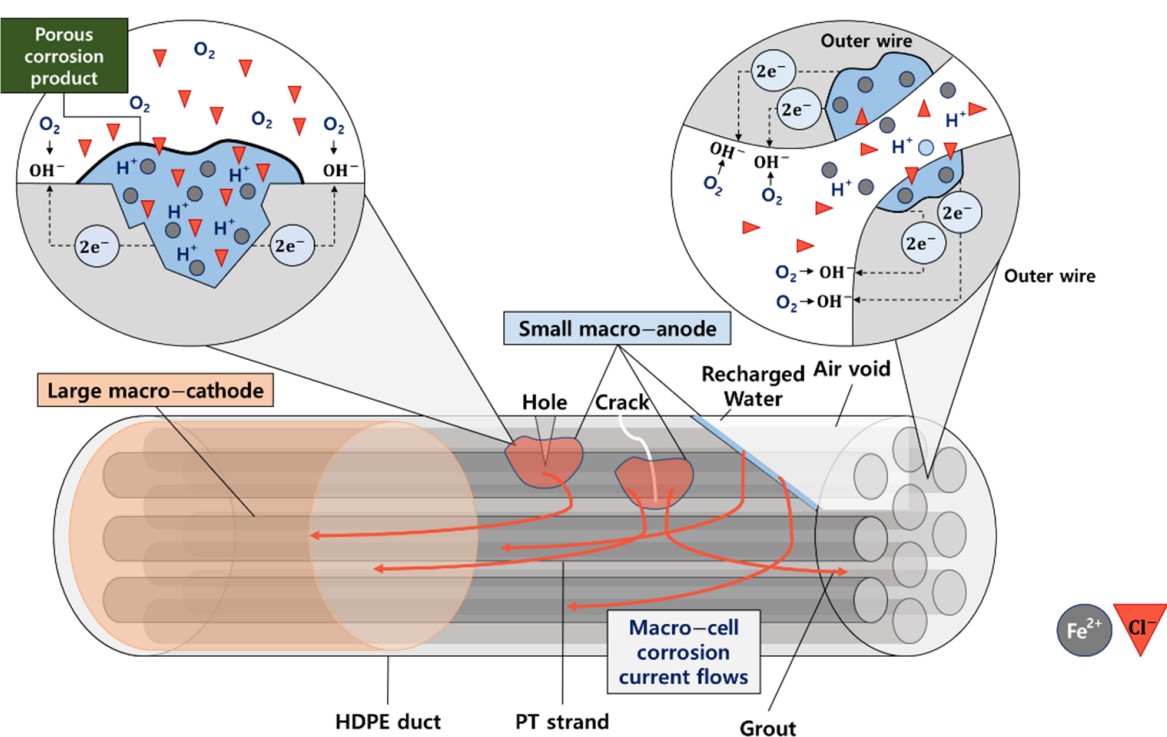

**Figure 1.** Corrosion mechanism of pitting, crevice, and macro-corrosion.

### 2.2. Corrosion Detection Sensor

As described in Section 1, many non-destructive devices have been developed for evaluating the corrosion of reinforcement in RC structures. In this section, a replaceable OCP measuring sensor is described. In this study, agar was used as the ion transfer material of the replaceable sensor. Agar, which is a material widely used for cell culture, is almost insoluble in organic solvents and has high resistance to chemicals. It can absorb a large amount of water and retain its gel condition when cooled at room temperature. Based on the results of previous studies [25–27], cement paste was prepared by mixing water and cement at a 1:4 weight ratio and adding a water-reducing agent, and then it was attached to the bottom of the socket. Agar was added into a prepared 1 M $KNO_3$ aqueous solution at a 2% weight ratio, and melted for 60 min at 120 °C to be used as an ion-exchange membrane.

The solution was then cooled to approximately 70~80 °C and poured onto hardened cement paste in the mold at a thickness of 4.0 ± 1.0 mm. It was cooled and hardened at room temperature for 12 h before the test.

After injecting the working solution into the socket, RE was inserted for use. Agar is a well-proven material as a salt-bridge in electrochemistry research, so it was utilized not only for salt-bridges but also for membranes for acquiring electrochemical connection between concrete and the electrolyte for our reference electrode. Many other researchers have reported the reliability of agar as salt-bridges [25,28,29]. In this study, Hg/HgO electrodes and a 1 M NaOH aqueous solution were used as the medium solution. Figure 2 shows photos of the sensor fabrication process and OCP measurement.

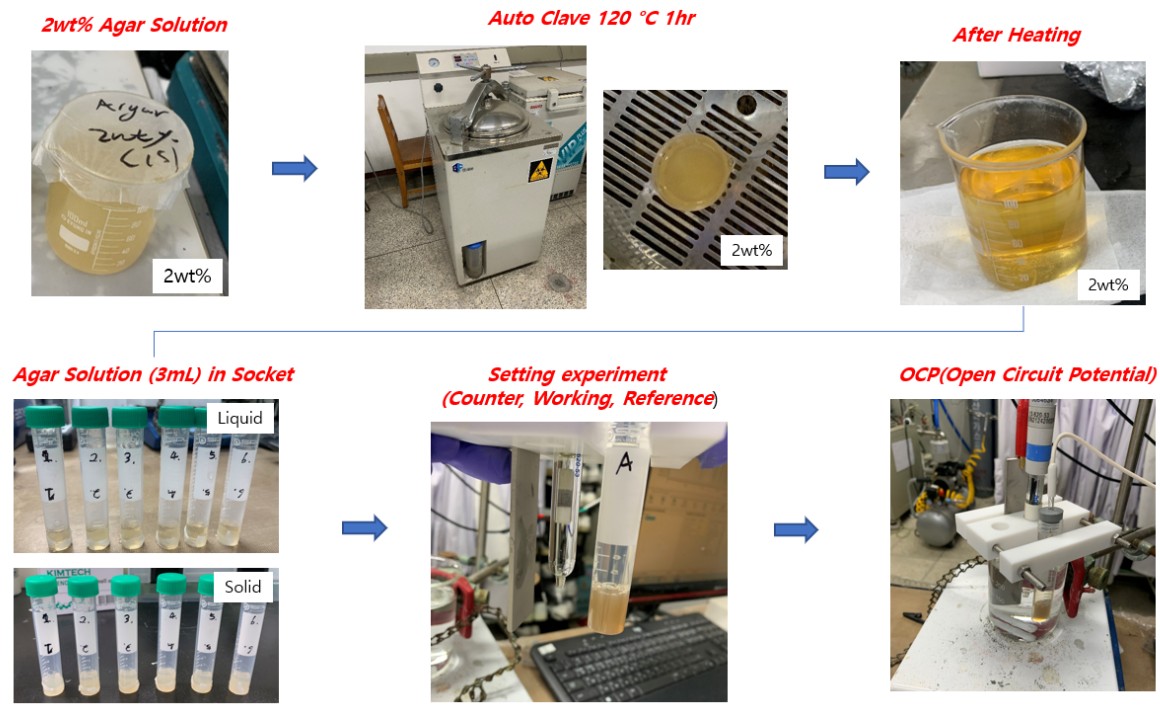

**Figure 2.** Fabrication process and photos for the agar sensor platform and OCP measurement.

### 3. Test Program for Accelerated Chloride Ingress and Potential Measurement

*3.1. RC Samples and Used Material*

3.1.1. Fabricated RC Samples with Different Cover Depths and *w/c* Ratios

For two years of a long-term monitoring test, cuboidal RC molds in a size of 300 mm × 200 mm × 130 mm were fabricated. The cover depth of the RC sample was set to 30, 45, and 60 mm. To induce corrosion in the central part, both ends of the steel were coated with epoxy except for 150 mm in the middle part of the reinforcement. SUS mesh was installed at 5 mm from the embedded steel reinforcement to measure the OCP, and the position of the socket-type agar sensor was adjusted at the same distance from the steel and SUS mesh. The SUS mesh had a shape of 30 × 30 mm and was made of the wire with a diameter of 0.5 mm. The agar socket had a 15 mm diameter and 100 mm length. For preventing the temperature effect on corrosion behavior, the specimens were tested at room temperature (21 °C ~ 25 °C). Wires were connected to the steel reinforcement acting as the WE and to the SUS acting as the CE, and then soldering and an epoxy coating were applied. This process allowed the electrodes to be connected from the outside of the sample before concrete pouring. Figure 3 shows a photograph of the mold and internal wire connection. The RC sample geometry and installation photos are shown in Figure 4. As previously explained, the agar-based socket can act as a salt bridge and can be inserted with a real reference electrode. The replaceable agar sensors are shown in the figures below (red circles).

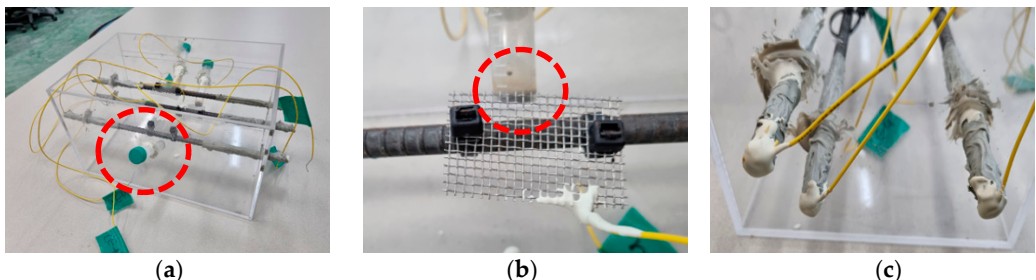

**Figure 3.** Reinforced concrete (RC) samples and wire connection for open-circuit potential (OCP) measurement. (**a**) Photo of RC samples; wire connection for (**b**) counter electrode (CE) and (**c**) working electrode (WE).

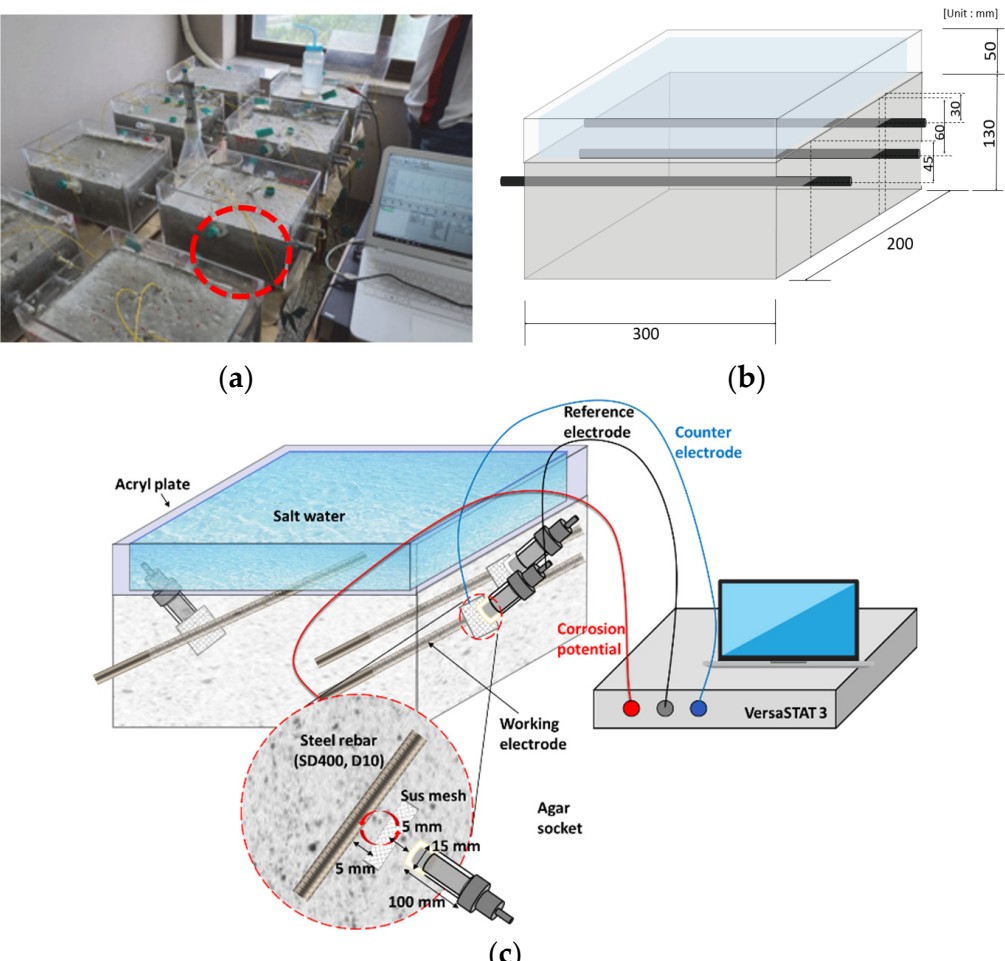

**Figure 4.** Photo and schematic diagram for RC samples for the test. (**a**) Photo of cured RC samples; (**b**) geometry of RC samples and (**c**) agar sensor installation and measurement connection.

### 3.1.2. Mix Proportions and Materials

OPC was used to prepare RC samples, and mix proportions with an air content of 4.5%, $w/c$ ratio of 0.4–0.6, and a flow of 600.0 mm were prepared. Table 1 shows the concrete mix proportions and the physical/chemical properties of cement. Tables 2 and 3 show the properties of cement and aggregates, respectively. The information of steel reinforcement is summarized in Table 4.

**Table 1.** Mix proportions for the test.

| No. | W/C | S/a (%) | Unit Weight (kg/m$^3$) | | | | |
|-----|-----|---------|-------|--------|------|--------|------------|
| | | | Water | Cement | Sand | Gravel | Admixtures |
| 1 | 0.4 | 43.0 | 180 | 450 | 712 | 966 | 3.15 |
| 2 | 0.5 | 45.0 | 180 | 360 | 779 | 974 | 2.52 |
| 3 | 0.6 | 47.0 | 180 | 300 | 837 | 966 | 2.10 |

**Table 2.** Chemical and physical properties of ordinary Portland cement (OPC).

| Items <br> Type | Chemical Compositions (%) | | | | | | | Physical Properties | |
|------|---------|-----------|-----------|-----|-----|-----|----------|------------------|-----------------|
| | $SiO_2$ | $Al_2O_3$ | $Fe_2O_3$ | CaO | MgO | $SO_3$ | Ig. Loss | Specific Gravity | Blaine (cm$^2$/g) |
| OPC | 21.96 | 5.27 | 3.44 | 63.41 | 2.13 | 1.96 | 0.79 | 3.16 | 3214 |

**Table 3.** Physical properties of fine and coarse aggregates.

| Items <br> Type | $G_{max}$ (mm) | Specific Gravity (g/cm$^3$) | Absorption (%) | F.M |
|------|-------------|------------------|------------|-----|
| Fine aggregate | - | 2.62 | 1.01 | 2.90 |
| Coarse aggregate | 20 | 2.68 | 0.82 | 6.87 |

**Table 4.** Propertied of the used reinforcement.

| Items <br> Type | Diameter (mm) | Yield Stength (MPa) | Steel Ratio | Spacing (mm) |
|------|-----------|----------------|-------------|-----------|
| Steel rebar | 10 | 400 | 0.90 | 41.5 |

As shown in Figure 1, a reinforcement bundle or tendons in a relatively small area may have complicated corrosion behavior such as a microcell, crevice cell, and macro cell. In some subway structure, corrosion in buried steel due to stray current has been reported [30]. If the steel spacing between the upper and lower steel is small, chloride condensation and a different chloride distribution can be caused [31]. In the test, the steel ratio and minimum spacing were 0.90% and 41.5 mm, respectively, so the independent corrosion in each steel was monitored.

### 3.2. Accelerated Chloride Intrusion Test and OCP Measurement

Regarding the saltwater for accelerating the corrosion of steel, three different chloride concentrations (0.0%, 3.5%, and 7.0% of wt.) were adopted. The 3.5% concentration is the ordinary level of sea water. Higher levels of chloride concentration have been reported when deicing agent ($CaCl_2$) is used for concrete pavements, so a 7.0% concentration was additionally considered. Another reason was that three levels are at least required for non-linear fitting for the evaluation of chloride behavior. One-way penetration from the top of the RC sample was induced. For long-term monitoring, dry-air curing was performed for four weeks after concrete pouring, and then an accelerated corrosion test through saltwater was conducted. Artificial seawater with three concentrations was left on the top for two weeks and then removed. Subsequently, the OCP under the wet condition was measured. Afterward, the dry condition was maintained for two weeks, and the OCP under the dry condition was measured again. Two weeks (wet) in the presence of saltwater and two weeks (dry) in the absence of saltwater were set to one cycle, and two cycles (eight weeks) were performed. After testing for two cycles (eight weeks), the measurement cycle was changed. The OCP measurement under the wet condition after the presence of saltwater

for six weeks and that under the dry condition after the absence of saltwater for two weeks were set to one cycle, and the test was conducted. Figure 5 shows the cyclic drying–wetting test process for two years. A potentiostat (VERSASTAT3, AMTEK) was used as the OCP measuring equipment. Figure 6 shows photographs of the cyclic accelerated test and OCP measurement.

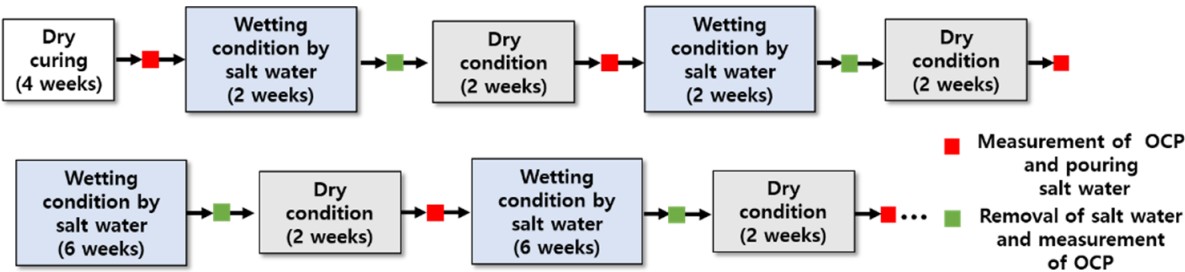

**Figure 5.** Accelerated corrosive conditions for the corrosion monitoring test.

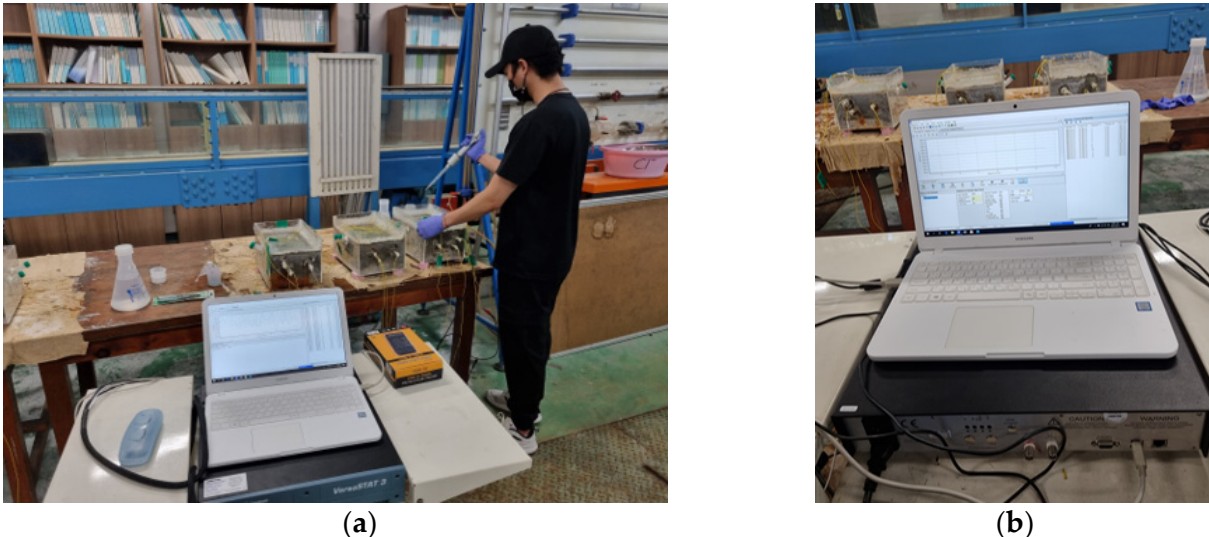

| (**a**) | (**b**) |

**Figure 6.** Open-circuit potential (OCP) monitoring stage and equipment. (**a**) OCP monitoring during accelerating conditions; (**b**) OCP measurement equipment.

## 4. Corrosion Potential Behavior with Test Parameters

### 4.1. OCP Evaluation with the Averaged OCP

The OCP measurement for two years is shown in Figures 7–9 with increasing chloride concentration. Although data fluctuation occurred due to the non-homogeneity of concrete, the OCP value increased in the negative direction with higher chloride concentration. Some OCP values were recovered due to the changes in temperature and local saturation condition. In particular, when corrosion occurs, the OCP value recovers or the corrosion rate decreases during corrosion progress [32]. This is because the corrosion product (rust) generated locally on the surface hinders the further oxidation of steel reinforcement. When long-term measurement is performed, the OCP decreases again in a direction that propagates corrosion. In addition, the dry condition was maintained before exposing the specimens to saltwater for increasing the inflow of oxygen. As the OCP values measured during this period showed large fluctuations due to a reduction in potential, the average values of three samples were used after data filtering.

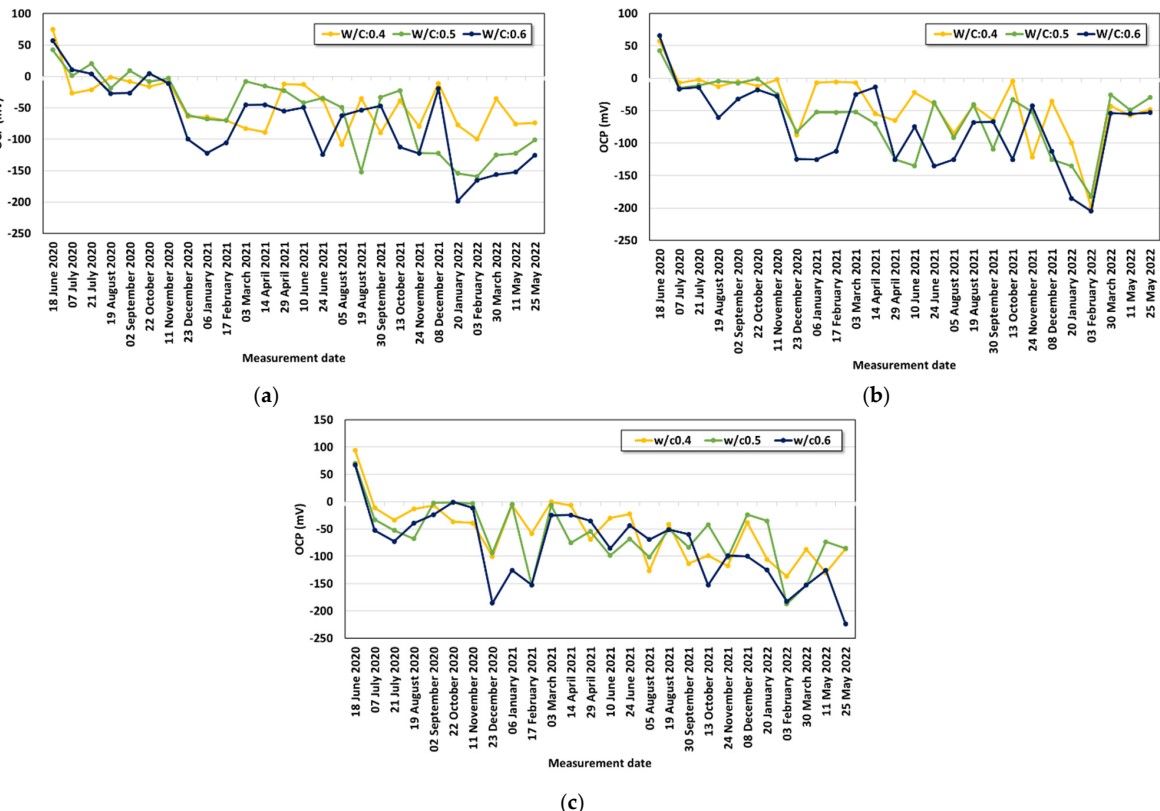

**Figure 7.** Open-circuit potential (OCP) evaluation under 0.0% chloride exposure conditions. Cover depths: (**a**) 60 mm, (**b**) 45 mm, and (**c**) 30 mm.

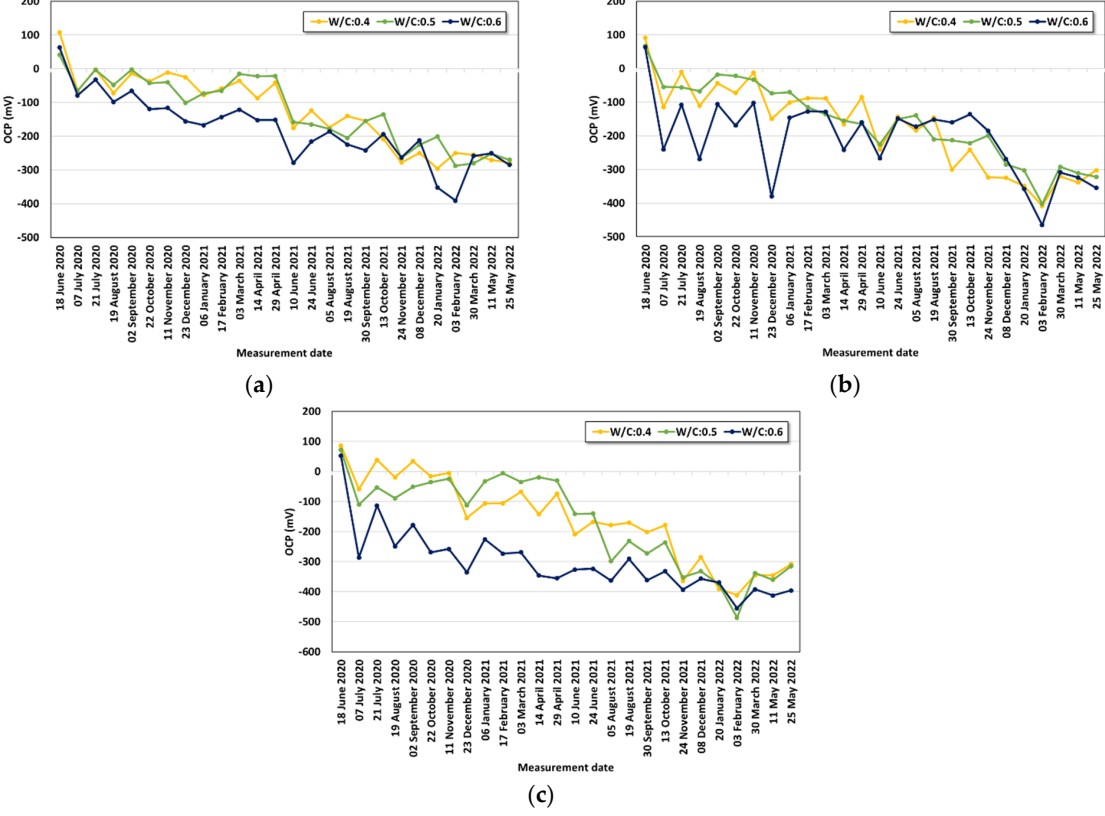

**Figure 8.** Open-circuit potential (OCP) evaluation under 3.5% chloride exposure conditions. Cover depths: (**a**) 60 mm, (**b**) 45 mm, and (**c**) 30 mm.

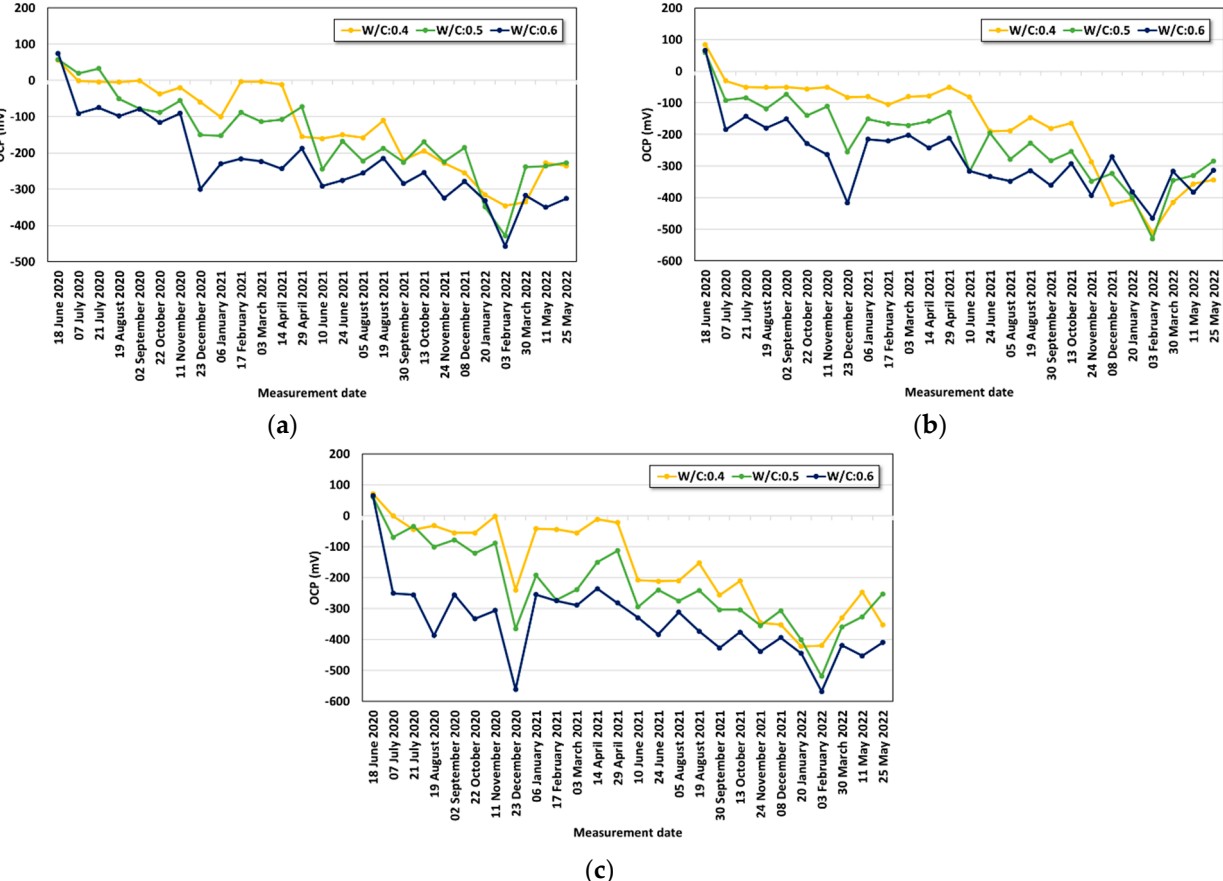

**Figure 9.** Open-circuit potential (OCP) evaluation under 7.0% chloride exposure conditions. Cover depths: (**a**) 60 mm, (**b**) 45 mm, and (**c**) 30 mm.

The critical corrosion potential of the used electrode was −450 mV [33], but the OCP that meets the value was measured from a cover depth of 30 mm after one year at a chloride concentration of 7.0% regardless of the $w/c$ ratio. After one year of corrosion monitoring, some specimens were dismantled, and the steel reinforcement in the corrosion-induced part was collected. The rust on the surface was then removed using citric acid. Afterward, the surface condition of the steel was examined using a microscope (DEM-MUT) with 10 times magnification. For the sample that contained no chloride in the aqueous solution, the degree of corrosion was insignificant. In the two cases with high chloride concentrations (3.5% and 7.0%), however, significant pitting-type corrosion was evaluated. Figure 10 shows the surface corrosion condition after one year.

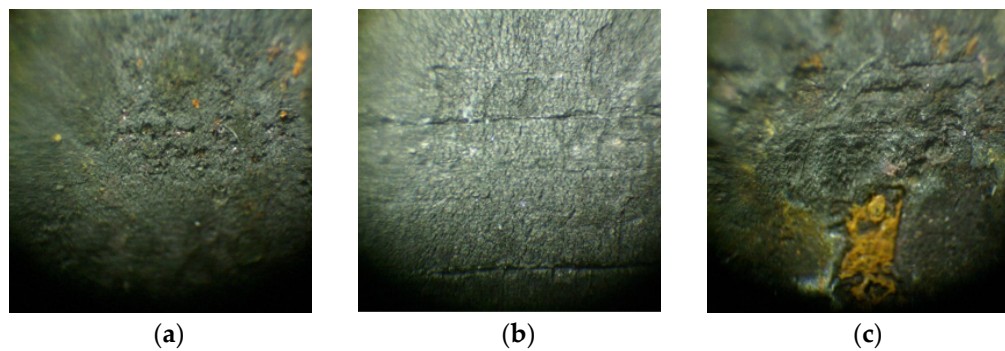

**Figure 10.** Photos of the corroded area with increasing chloride concentration after 1 year: (**a**) 0.0%, (**b**) 3.0%, and (**c**) 7.0% (cover depth: 45 mm, $w/c$: 60%).

In this study, the progress of pitting corrosion with increasing chloride concentration was only observed, and the corrosion weight and area were not evaluated. Unlike the corrosion pattern due to carbonation, local corrosion pits are caused by chloride penetration and are accelerated in the cracked area [34]. In the previous studies [35,36], the flexural capacity of an RC beam was evaluated considering the pitting corrosion effect. The residual steel area and distribution function of the corrosion pit were modeled. The probability of failure was evaluated with the pit interference effect, which showed significant increases in corrosion probability at the initial uniform corrosion condition. After the 2-year test, the agar material showed a stable form without changing color and swelling. The agar-melting process for 120 min and distilled water for cement mixing with high alkalinity were effective for bacteria growth control. Careful investigation of preventing bio-deterioration from bacteria is required for the long-term test over 2 years and mass production of the agar-based sensor. The reproducibility test after the corrosion test was performed for 150 s with 1 M $KNO_3$ and a reference of $Ag/AgCl$. The average of the measured results from 6 samples showed that 0.05208 mV (OCP) and 0.342 of COV were evaluated.

*4.2. Relationship between OCP and the Parameters*

4.2.1. Effects of the Cover Depth and $w/c$ Ratio on the OCP with Increasing Chloride Content

It is difficult to quantify the OCP values at each measurement point by comparing them with the factors affecting corrosion ($w/c$, cover depth, and chloride concentration). The OCP change tendency according to the factors can be clearly identified by averaging the OCP values measured for two years. Figure 11 shows the average values with increasing cover depth. The corrosion potential decreased as the cover depth increased and the $w/c$ ratio decreased. When the OCP ratio was analyzed based on a corrosion potential of −450 mV, it was found to be approximately 18.4% even under the most unfavorable conditions (30 mm, $w/c$ 0.6) at a chloride concentration of 0.0%. At 3.5%, however, it significantly increased to 40.1 ~ 67.4% for $w/c$ 0.6, 27.6~37.7% for $w/c$ 0.5, and 28.0~39.1% for $w/c$ 0.4. This tendency increased from ~49.8 to ~76.5%, 33.8% to 50.8%, and 28.0% to 37.3%, respectively, at a chloride concentration of 7.0%.

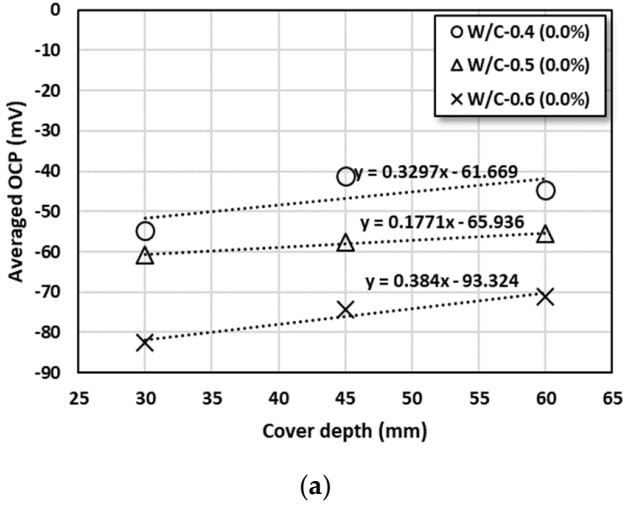

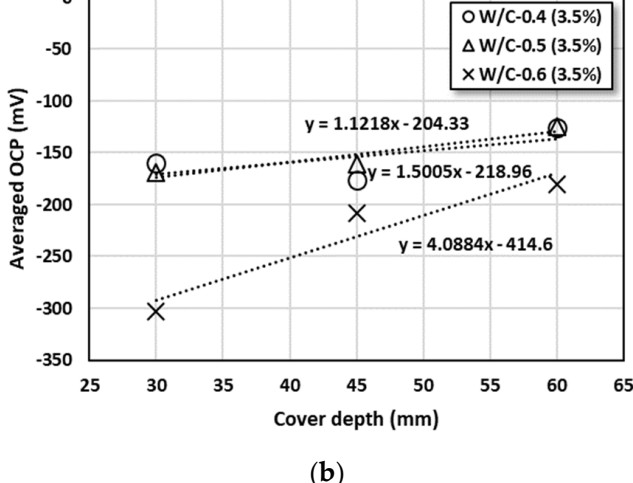

(a)                                                     (b)

**Figure 11.** *Cont.*

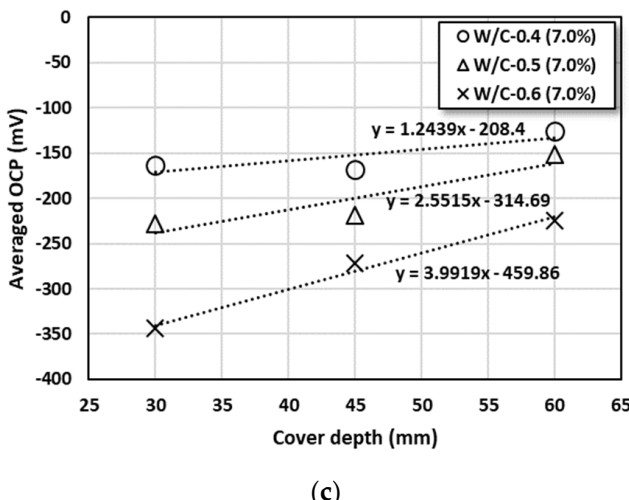

(**c**)

**Figure 11.** Averaged open-circuit potential (OCP) with increasing chloride concentration: (**a**) 0.0%, (**b**) 3.5%, and (**c**) 7.0%.

When each average OCP value was evaluated through linear regression analysis, the gradient can be assumed as the degree to affect the OCP. When the x-axis (cover depth) in Figure 11 was subjected to linear regression analysis with respect to the OCP, the gradient can be summarized as shown in Table 5. As can be seen from the table, the OCP recovery through the cover depth was rapidly improved in all cases as the $w/c$ ratio increased. This tendency indicates that securing the cover depth is an important protection mechanism that can reduce the corrosion risk in the condition of low-quality concrete and high external chloride concentration. The results can provide useful information for durability design such as determining the minimum cover depth and maximum $w/c$ ratio of concrete. A critical chloride content inducing corrosion is usually adopted for service life evaluation; however, it is only assumed, and the conservative index since corrosion initiation occurs very complicatedly and the value varies a lot with experimental conditions [1,2,6].

**Table 5.** Gradient analysis of the cover depth effect on the open-circuit potential (OCP).

| | Averaged OCP = a × Cover Depth + b | | |
|---|---|---|---|
| | **a** | **b** | **$R^2$** |
| WC-0.4 (0.0%) | 0.3297 | −61.669 | 0.5075 |
| WC-0.5 (0.0%) | 0.1771 | −65.936 | 0.9926 |
| WC-0.6 (0.0%) | 0.384 | −93.324 | 0.9377 |
| WC-0.4 (3.5%) | 1.1218 | −204.33 | 0.4354 |
| WC-0.5 (3.5%) | 1.5005 | −218.96 | 0.8921 |
| WC-0.6 (3.5%) | 4.0884 | −414.6 | 0.9099 |
| WC-0.4 (7.0%) | 1.2439 | −208.4 | 0.6597 |
| WC-0.5 (7.0%) | 2.5515 | −314.69 | 0.8382 |
| WC-0.6 (7.0%) | 3.9919 | −459.86 | 0.987 |

### 4.2.2. Effects of the Cover Depth and $w/c$ Ratio on the OCP with Increasing Period

Owing to the periodic infiltration of saltwater, the chloride content increases inside the concrete, and it yields a condition vulnerable to corrosion despite the continuous cement hydration. In Figure 12, the OCP values for two years were classified into four stages to analyze the OCP value with the exposure period. When the $w/c$ ratio was 0.4, a significant drop in the OCP value began to be observed under 3.5% and 7.0% conditions only after one year. When the $w/c$ ratio was higher ($w/c$: 0.6), values below −100 mV were observed

regardless of the cover depth after six months. At cover depths of 30 and 45 mm, the OCP linearly and significantly decreased and reached the critical level of −450 mV. The OCP reduction over time was mainly caused by the inflow of moisture and oxygen under dry and wet conditions. A rapid reduction in the OCP was measured in the saltwater environment that exceeded the seawater condition (7.0%).

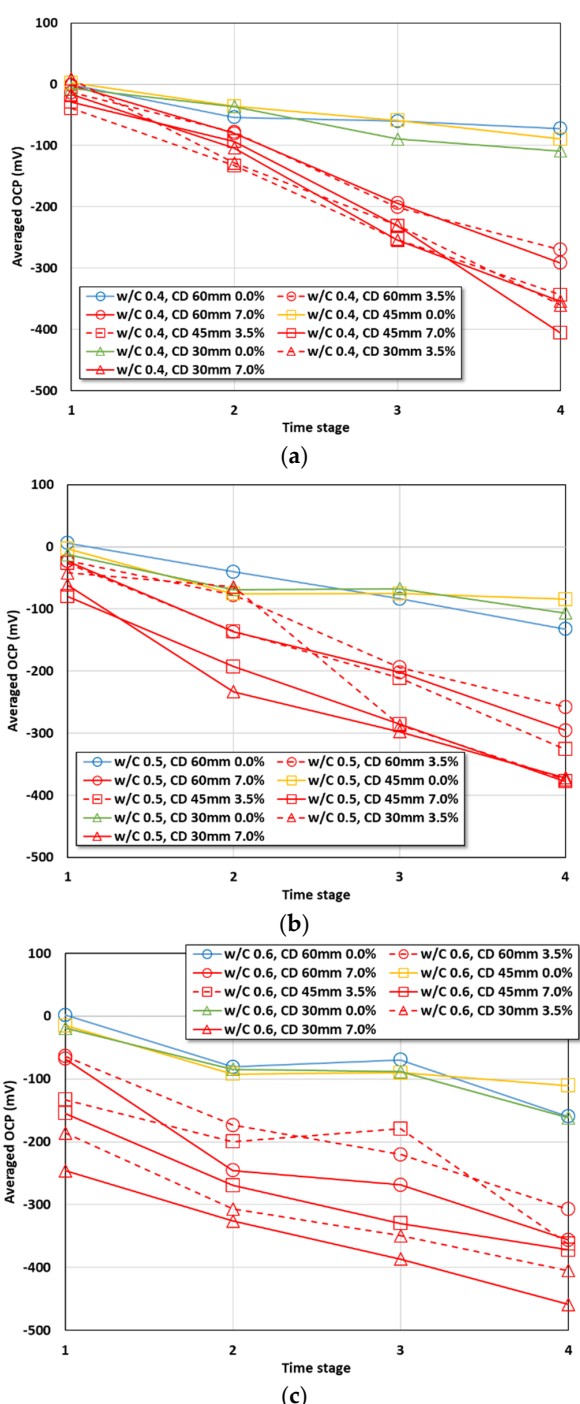

**Figure 12.** Open-circuit potential (OCP) drop with increasing exposure period. Water-to-cement (*w/c*) ratios: (**a**) 0.4, (**b**) 0.5, and (**c**) 0.6.

When each exposure environment is fixed, a relatively high coefficient of determination can be derived by obtaining multiple correlations of time, cover depth, and OCP. Figure 13 shows the OCP correlations for the seawater concentration (3.5%) condition and the severe seawater condition (7.0%). A higher correlation was derived as the chloride concentration

increased, and each result was shown as a contour and a regression equation. A weak correlation was observed under the wet condition (0.0% concentration), but the correlation was improved and the corrosion potential decreased with high chloride concentration. From the wet condition, as the chloride concentration increased to 3.5% and 7.0%, the multiple correlation coefficient increased from 0.627 to 0.634 (0.0%), 0.789 to 0.905 (3.5%), and 0.783 to 0.878 (7.0%), respectively. The coefficient of determination also increased from 0.614 to 0.771 under the 7.0% condition. The regression equations under each condition are summarized in Table 6, and the OCP contour for each condition is shown in Figure 14. It is well reported that the exposure condition is very important as the exterior chloride concentration governs the surface chloride content, and it determines the required cover depth and concrete properties for the intended service life [1,6,37]. Initial chloride ions from sea sand or chemical admixtures usually remain constant without the dissociation of bound chlorides due to the pH drop [37,38]. The surface chloride content increases with the exposure period and remains constant after 20~30 years [39,40]. The results from the test showed a clear relationship with longer exposure period and higher chloride content, and they were directly compared with averaged OCP, which meant actual corrosion behavior. The linear OCP relation with cover depth can be used for a referential guide for durability design.

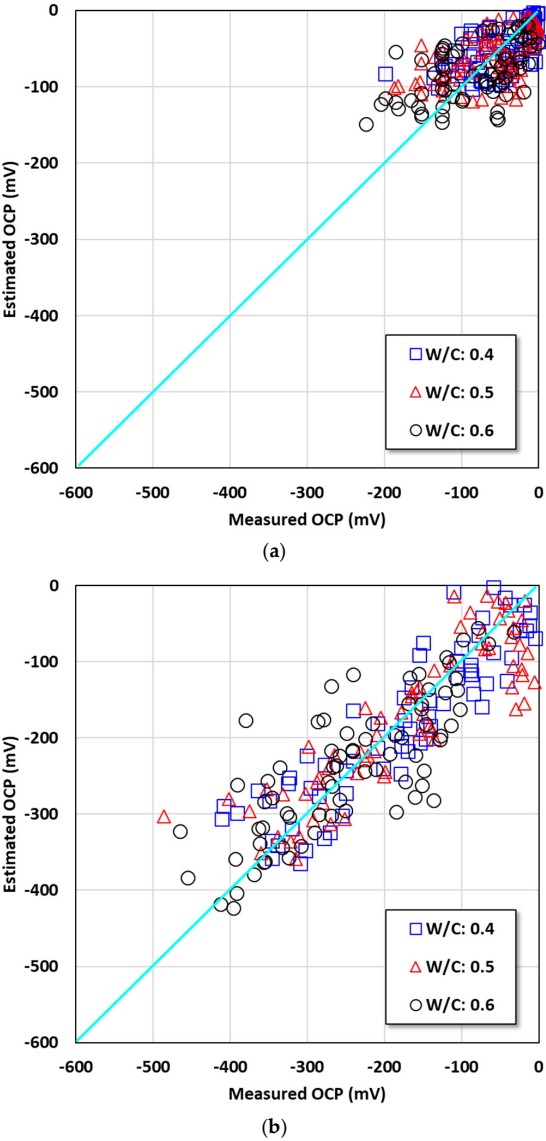

**Figure 13.** *Cont.*

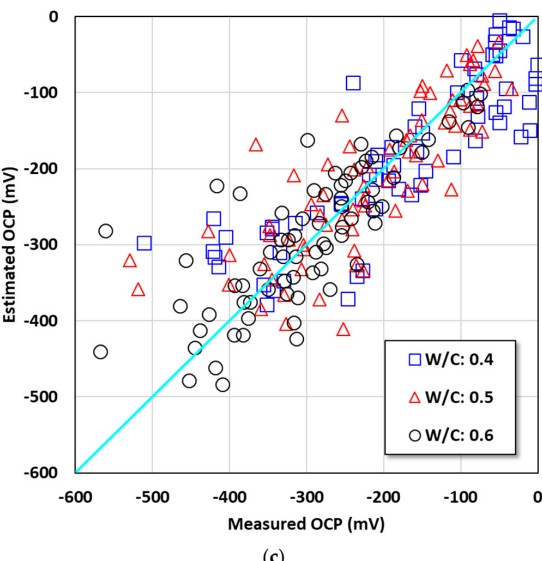

(**c**)

**Figure 13.** Regression analysis of the corrosion parameters for open-circuit potential (OCP): (**a**) wet condition, (**b**) 3.5% concentration, and (**c**) 7.5% concentration.

**Table 6.** Regression results of the open-circuit potential (OCP) with exposure and cover depth. *W/C*, water-to-cement.

| Condition | W/C Ratio | OCP = A(Time) + B(Cover Depth) + C | | | | |
|---|---|---|---|---|---|---|
| | | Multiple Correlation | Determination Coefficient | A | B | C |
| **Wet Condition** | 0.4 | 0.634 | 0.402 | −0.140 | 0.330 | −14.392 |
| | 0.5 | 0.627 | 0.393 | −0.159 | 0.177 | −12.385 |
| | 0.6 | 0.629 | 0.395 | −0.183 | 0.384 | −31.730 |
| **3.5% Concentration** | 0.4 | 0.905 | 0.819 | −0.528 | 1.122 | −26.206 |
| | 0.5 | 0.887 | 0.787 | −0.501 | 1.501 | −49.861 |
| | 0.6 | 0.789 | 0.622 | −0.357 | 4.088 | −294.175 |
| **7.0% Concentration** | 0.4 | 0.878 | 0.771 | −0.564 | 1.244 | −18.187 |
| | 0.5 | 0.838 | 0.702 | −0.468 | 2.552 | −156.681 |
| | 0.6 | 0.783 | 0.614 | −0.390 | 3.992 | −328.259 |

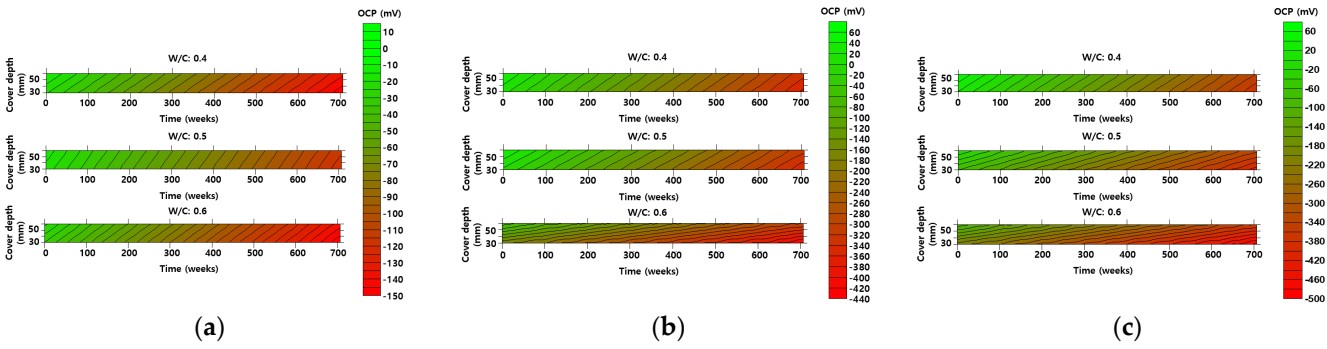

(**a**)       (**b**)       (**c**)

**Figure 14.** Open-circuit potential (OCP) contour with corrosion parameters: (**a**) wet condition, (**b**) 3.5% concentration, and (**c**) 7.5% concentration.

## 5. Conclusions

In this study, reinforced concrete (RC) samples were prepared considering the chloride concentration, cover depth, and water-to-cement ($w/c$) ratio. In addition, the open-circuit potential (OCP) of the embedded steel reinforcement was measured by conducting a cyclic drying–wetting test in a saltwater environment for two years, and its relationships were obtained considering durability design parameters and exposure period. The conclusions and research limitation drawn in this study are as follows:

(1) Under the conditions that considered saltwater, the OCP was evaluated to be lower than the critical potential ($-450$ mV) regardless of the $w/c$ ratio under the lowest cover depth condition (30 mm) after six months. As corrosion progressed, the OCP varied due to the partial saturation and rust product.

(2) The averaged OCP measured for two years was derived, and each average value was analyzed with the cover depth. The corrosion potential decreased with increasing chloride exposure period and high chloride concentration. When the $w/c$ ratio was 0.4, a significant drop in the OCP value began to be observed under the 3.5% and 7.0% conditions only after one year. Under the condition with a high $w/c$ ratio ($w/c$: 0.6), values below $-100$ mV were observed regardless of the cover depth after six months. The averaged OCP increased linearly in the negative direction with decreasing cover depth, and they showed a clearer tendency with higher chloride concentration. Regression equations were evaluated for the OCP behavior for each exposure environment considering the exposure period and cover depth.

(3) The measured OCP showed fluctuations with varying saturation and the corrosion pitting effect was not investigated. If the bearing capacity in the corroded RC beam with the mineral admixture such as fly ash and slag was evaluated through long-term accelerated conditions, the results from this work would show more reasonable ones that can be applied to a real RC member.

**Author Contributions:** Conceptualization, S.-J.K. and B.-Y.Y.; methodology, S.-J.K. and K.-M.L.; formal analysis, S.-J.K. and Y.-S.Y.; investigation, S.-J.K. and B.-Y.Y.; resources, S.-J.K. and K.-M.L.; data curation, S.-J.K. and K.-H.Y.; writing—original draft preparation, S.-J.K. and Y.-S.Y.; writing—review and editing, S.-J.K. and Y.-S.Y.; supervision, K.-H.Y. and K.-M.L.; funding acquisition, S.-J.K. and K.-M.L. All authors have read and agreed to the published version of the manuscript.

**Funding:** This research was supported by the National Research Foundation of Korea (NRF-No. NRF-2020R1A2C2009462) and the Korea Agency for Infrastructure Technology Advancement (KAIA) grant funded by the Ministry of Land, Infrastructure, and Transport (Grant 22SCIP-C158977-03).

**Institutional Review Board Statement:** Not applicable.

**Informed Consent Statement:** Not applicable.

**Data Availability Statement:** The data presented in this study are available on request from the corresponding author.

**Conflicts of Interest:** The authors declare no conflict of interest.

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
