# Peer review of "Corrosion Behavior in RC Member with Different Cover Depths under Cyclic Chloride Ingress Conditions for 2 Years"

_applsci, doi:10.3390/app122413002_

Round 1
Reviewer 1 Report
The reviewed paper entitled “Corrosion behavior in RC member with different cover depths 2 under cyclic chloride ingress conditions for 2 years” deals with the corrosion behavior of reinforced concrete under chloride solution exposure.
Although the paper has some potential elements that can increase the degree of novelty, they are not appropriately explored. In this current form I do not recommend the paper for publishing.
Mainly, the authors use a solid electrolyte which is novel, but insufficiently well described. The agar’s reliability, improvement or any possible source of interference with the measurements are not scrutinized based on the relevant literature. Also, the improvements or limitations of agar with respect to classical methods are not highlighted based on the current state of the art (with reference to any standard for corrosion of reinforced steel and/or relevant literature).
Authors suggest that the sensor can be embedded, replaced, and calibrated but fail to address these important issues in the paper, which is a large loss in the potential impact of the paper.
I consider that after a targeted focus on the strengths given by the novelty, the work has more potential to have an impact on the relevant fields of study. Due to the potential of the novel aproach, I thus consider that the authors can re-submit the paper after appropriate restructuring.
Author Response
Reviewer 1
The reviewed paper entitled “Corrosion behavior in RC member with different cover depths 2 under cyclic chloride ingress conditions for 2 years” deals with the corrosion behavior of reinforced concrete under chloride solution exposure.
Although the paper has some potential elements that can increase the degree of novelty, they are not appropriately explored. In this current form I do not recommend the paper for publishing.
Q1] Mainly, the authors use a solid electrolyte which is novel, but insufficiently well described. The agar’s reliability, improvement or any possible source of interference with the measurements are not scrutinized based on the relevant literature. Also, the improvements or limitations of agar with respect to classical methods are not highlighted based on the current state of the art (with reference to any standard for corrosion of reinforced steel and/or relevant literature).
A1] Aga is well-proven materials as a salt-bridge in electrochemistry research, and we utilize this not only salt-bridge but also membrane for acquiring electrochemical connection between concrete and the electrolyte for our reference electrode. Many other researchers report the reliability of agar as a salt-bridges and we added references as following: The manuscript was enhanced in [Section 2.2, line 117-138, line 287-292]
Additional references)
- Hassel, A.W.; Fushimi, K.; Seo, M. An agar-based silver| silver chloride reference electrode for use in micro-electrochemistry. Electrochem. Commun. 1999, 1, pp.180-183.
- Shao, X.M.; Feldman, J.L. Micro-agar salt bridge in patch-clamp electrode holder stabilizes electrode potentials. J. neurosci. methods , 159, pp.108-115.
- Kakiuchi, T. Salt bridge in electroanalytical chemistry: Past, present, and future. J. Solid State Electrochem. 2011, 15, pp.1661-1671.
Q2] Authors suggest that the sensor can be embedded, replaced, and calibrated but fail to address these important issues in the paper, which is a large loss in the potential impact of the paper.
A2] As you see the schematics of our sensor installation, we simply embedded socket which contains reference electrolyte and electrochemically connected to concrete through agar salt bridge. Real reference electrode, which is working as a sensor for corrosion potential as well as current is only placed in the socket when user want to measure it. Therefore, we suggested our sensor is available for embedding, replacing as well as calibrating. The research highlight, limitation, and originality were enhanced in [Introduction and conclusions, line 55-89 and line 341-361]
I consider that after a targeted focus on the strengths given by the novelty, the work has more potential to have an impact on the relevant fields of study. Due to the potential of the novel approach, I thus consider that the authors can re-submit the paper after appropriate restructuring

Reviewer 2 Report
The authors conducted cyclic drying–wetting experimental tests to study the corrosion potential by considering the chloride concentration, water-to-cement (w/c) ratio, and cover depth at three levels. This paper is expected to be considered for publication in the journal if the authors address the following issues satisfactorily.
1. The contribution of this paper is limited. The research purpose and innovation are not very clear. Can you describe what is the main novelty of this work?
2. The introduction section and the literature review should be extended using all recent relevant research. The authors are suggested to add state-of-the-art references in the manuscripts.
3. Have you discussed the effect of accelerated corrosion rate on the results? How did you select the amount of the different chloride conventions?
4. You need to discuss the effect of pit sizes and locations on the results. Pitting corrosion can influence the results, and it is important to discuss this in the paper and the conclusion. I suggest reviewing the following papers:
o Kioumarsi, M.M., Hendriks, M.A., Kohler, J. and Geiker, M.R., 2016. The effect of interference of corrosion pits on the failure probability of a reinforced concrete beam. Engineering Structures, 114, pp.113-121.
o Kioumarsi, M., Markeset, G. and Hooshmandi, S., 2017. Effect of pit distance on failure probability of a corroded RC beam. Procedia Engineering, 171, pp.526-533.
5. The discussion section in the present version is relatively weak and should be strengthened with more details and justifications.
6. The authors must revise the conclusion and make it stronger. The current version does not reflect well the research significance. You do not need to repeat the percentages in the conclusion.
Author Response
The authors conducted cyclic drying–wetting experimental tests to study the corrosion potential by considering the chloride concentration, water-to-cement (w/c) ratio, and cover depth at three levels. This paper is expected to be considered for publication in the journal if the authors address the following issues satisfactorily.
Q1] The contribution of this paper is limited. The research purpose and innovation are not very clear. Can you describe what is the main novelty of this work?
A1] The objectives of the manuscript were an application of newly developed sensor which can be replaceable from outside and the corrosion behavior evaluation through comparison with the data from the adopted sensor and durability design parameters (exterior chloride content, cover depth, and w/c ratios). As commented, the research highlight was enhanced in introduction and conclusions. [line 73-89, and line 340-365]
Q2] The introduction section and the literature review should be extended using all recent relevant research. The authors are suggested to add state-of-the-art references in the manuscripts.
A2] As commented, many references were enhanced regarding electro-chemical sensors for corrosion detection. They were added in Introduction, but this paper is not a review paper and focused on corrosion behavior evaluation for long-term exposure test [line 49-71 in Chapter 1].
Q3] Have you discussed the effect of accelerated corrosion rate on the results? How did you select the amount of the different chloride conventions?
A3] It is very difficult to simulate the actual corrosive condition with accelerated test, even if using impressed current method. Conventionally 3.5% of chloride concentration shows average sea condition (0.50~0.52 mol/l), however surface chloride content of concrete is different from sea water since it depends on porosity, absorption rate of cement hydrates, and moisture content. Chloride condensation using deicing agent often shows higher than sea-level concentration, so that 7.0 % (two times higher than sea-level) was considered for the most severe level. Another reason was at least three levels were required for handling non-linear relationship (3 levels of w/c ratio, cover depth, and chloride concentration) (line 189-194, In 3.2 section).
Q4] You need to discuss the effect of pit sizes and locations on the results. Pitting corrosion can influence the results, and it is important to discuss this in the paper and the conclusion. I suggest reviewing the following papers:
A4] In this paper, a quantitative relation for durability design was obtained through long-term test. Service life of RC (Reinforced Concrete) structures is defined as the conditions for chloride attack: induced chloride content does not exceed to the critical chloride content at steel location within the intended service life. Regarding carbonation, durability design is performed assuming that the increasing carbonation depth does not exceed to cover depth within intended service life. Pitting corrosion is usually caused in the chloride ingress. In the work, pitting corrosion was observed but the weight loss or corrosion area were not measured. The two recommended references were added and well-explained in the revised version [Section 4.1]. [Line 251-258, line 362-365]
Q5] The discussion section in the present version is relatively weak and should be strengthened with more details and justifications.
A5] As commented, discussion part was enhanced. In 4.2.1, the effects of design parameters (cover depth and w/c ratio) were discussed with varying chloride content. In 4.2.2, they were discussed with increasing period. The trends of changing OCP were quantified through regression analysis and the results can provide useful information for durability design such as determination of cover depth and minimum w/c ratios. The related figure were improved with fitting line and the comparison were also enhanced. [line 287-291, Figure 10]
Q6] The authors must revise the conclusion and make it stronger. The current version does not reflect well the research significance. You do not need to repeat the percentages in the conclusion.
A6] The focus of this paper is obtaining a quantitative relationship among design parameters and corrosion behaviour from the sensor. Regarding the issue, introduction and conclusions were improved. The limitation of the work was clearly discussed. [Introduction/ Conclusions; line 80-89,line 340-365]
Reviewer 3 Report
Corrosion behavior in RC members with different cover depths is discussed imposing cyclic ingress conditions. This experimental research has been carried out for two years, which might be somewhat challenging, and provides interesting results. This reviewer believes this paper is well written and organized. On the other hand, more information about the specimen should be added to deepen the readers’ understanding as below.
1. The specimen size is mainly mentioned, but its details or design concept are not presented. Since the authors are emphasizing the unique point of this paper is focusing on the RC members, relevant photos and figures of the RC specimens employed should be shown and explained in detail.
2. Please explain the information about the reinforcement. Also, if possible, please explain the possible effects when the amount of the reinforcement is changed (Because the RC members have many reinforcing bars inside, not one bar inside as the specimen)
Author Response
Corrosion behavior in RC members with different cover depths is discussed imposing cyclic ingress conditions. This experimental research has been carried out for two years, which might be somewhat challenging, and provides interesting results. This reviewer believes this paper is well written and organized. On the other hand, more information about the specimen should be added to deepen the readers’ understanding as below.
Q1] The specimen size is mainly mentioned, but its details or design concept are not presented. Since the authors are emphasizing the unique point of this paper is focusing on the RC members, relevant photos and figures of the RC specimens employed should be shown and explained in detail.
A1] As commented, the schematic sketch and sample geometry were enhanced. [Figure 4. Line 154-158, In Chapter 3]
Q2] Please explain the information about the reinforcement. Also, if possible, please explain the possible effects when the amount of the reinforcement is changed (Because the RC members have many reinforcing bars inside, not one bar inside as the specimen)
A2] The used reinforcement has SD400 (fy=400MPa). In the test, the steel ratio and the spacing are 0.90 % and 41.5 mm, so that the spacing has enough to ignore the interaction of corrosion. The interaction effect of corrosion in contacted or near steel is explained. [in Section 3.1, line 173-181, Table 4]
Round 2
Reviewer 1 Report
The details provided in Fig. 4 b and c were of great contribution in increasing the quality of the electrochemical system presentation.
However, the author had not provided aspects that allow reproducibility with respect the experiments. Also, I still have concerns with respect to the reliability of agar in the 2 years experimental time frame.
Reproducibility:
* the agar gel conductivity should be provided, with at least four gel samples indentically prepared (standard error provided);
* the SUS sIeve mesh size, wire diameter and overall mesh area should be provided;
* the size of the agar socked (e.g. radius, height) should be provided;
* the distance from the mesh to the 10mm steel reinforcement should be drawn in fig 4 c;
* the room temperature (and it's variance) needs to be provided.
Reliability:
* the agar gel is a preferred medium for bacteria growth. What steps were taken to prevent bacterial growth?
Reviewer 2 Report
The authors addressed my comments, and modified the text accordingly. I recommend the publication of the paper in the current format.
Author Response
Really appreciate for your kind comments.
We authors treated the revised version more scientifically based on the another revier's comment